# A Simplified Coastline Inflection Method for Correcting Geolocation Errors in FengYun-3D Microwave Radiation Imager Images

Zhuoqi Chen [1,2,3], Jin Xie [4], Georg Heygster [5], Zhaohui Chi [6], Lei Yang [7], Shengli Wu [7], Fengming Hui [1,2,3] and Xiao Cheng [1,2,3,*]

1   School of Geospatial Engineering and Science, Sun Yat-sen University and Southern Marine Science and Engineering Guangdong Laboratory (Zhuhai), Zhuhai 519082, China
2   Key Laboratory of Natural Resources Monitoring in Tropical and Subtropical Area of South China, Ministry of Natural Resources, Zhuhai 519082, China
3   University Corporation for Polar Research, Zhuhai 519082, China
4   College of Global Change and Earth System Science, Beijing Normal University, Beijing 100875, China
5   GEORG-Lab (Geophysical Remote Sensing Lab), 28209 Bremen, Germany
6   Department of Geography, Texas A&M University, College Station, TX 77843, USA
7   National Satellite Meteorological Center, China Meteorological Administration, Beijing 100081, China
*   Correspondence: chengxiao9@mail.sysu.edu.cn; Tel.: +86-186-010-01669

**Abstract:** Passive microwave (PMW) sensors are popularly applied to Earth observations. However, the satellite PMW radiometer data sometimes have non-negligible errors in geolocation. Coastline inflection methods (CIMs) are widely used to improve geolocation errors of PMW images. However, they commonly require accuracy satellite flight parameters, which are difficult to obtain by users. In this study, a simplified coastline inflection method (SCIM) is proposed to correct the geolocation errors without demanding for the satellite flight parameters. SCIM is applied to improve geolocation errors of FengYun-3D (FY-3D) Microwave Radiation Imager (MWRI) brightness temperature images from 2018 and 2019. It reduces the geolocation errors of MWRI images to 0.15 pixels in the along-track and cross-track direction. This means reductions of 75% and 86% of the geolocation errors, respectively. The mean brightness temperature differences between the ascending and descending MWRI images are reduced by 34%, demonstrating the improved geolocation accuracy of SCIM. The corrected images are also used to estimate Arctic sea ice concentration (SIC). By comparing with SICs retrieved from the un-corrected images, the root mean square error (*RMSE*) and mean absolute error (MAE) of the SICs from the corrected images are reduced from 13.7% to 10.2% and 8.9% to 6.9%, respectively. The mean correlation coefficient (*R*) increases from 0.91 to 0.95. All these results indicate that SCIM can reduce geolocation errors of satellite-based PMW images significantly. As SCIM is very simple and easy to be applied, it could be a useful method for users of PMW images.

**Keywords:** simplified coastline inflection method (SCIM); passive microwave images; geolocation correction; sea ice concentration

## 1. Introduction

With the capability of the radiation to penetrate cloud cover, passive microwave (PMW) sensors are widely applied to Earth observations, such as soil moisture, land surface temperature, surface melt, etc. [1–4]. However, geolocation errors are often found in satellite PMW radiometer images. There exist 3.5 to 7 km geolocation errors in early versions of Advanced Microwave Scanning Radiometer-Earth Observing System (AMSR-E) images [5] and 1 pixel (about 25 km) of geolocation errors in Microwave Radiation Imager (MWRI) images [6]. The geolocation errors of Special Sensor Microwave/Imager (SSM/I) images were also found [7]. Since precise geolocation is fundamental to PMW data applications, the correction of geolocation errors is necessary for applications of PMW images [8].

Two types of methods for correcting geolocation errors of satellite images are proposed. The first, called the Image Correlation Method (ICM), uses correlation coefficients (*R*) of satellite images to correct the geolocation errors. It quantifies the similarity between an uncorrected image and a reference image with high geolocation accuracy. The ICM finds the best *R* between the uncorrected image and the reference image by varying the geolocation of the uncorrected image. The Normalized Cross Correlation (NCC) is a sub-type of method of ICMs, which is by definition the inverse Fourier transform of the convolution of the Fourier transform of two images. It is widely applied in the studies of image matching [9–11].

The second type of method improves geolocation errors of a satellite image by correcting satellite flight parameters, such as satellite orientation and observation angles etc. [12,13]. The node differential method (NDM) and the feature matching method (FMM) are included in this type of method. NDM detects geolocation errors by minimizing brightness temperature (BT) differences between ascending and descending PMW images. It assumes that most of the BT differences between ascending and descending PMW images obtained on the same day are caused by the geolocation errors [5,14]. FMM corrects the geolocation by matching ground control points (GCPs) in the uncorrected image with the corresponding points from a reference image [15]. A type of FMM, called the coastline inflection method (CIM), takes coastline inflection points as GCPs to correct the geolocation errors of PMW images [16,17].

Both approaches have their advantages and limitations. ICM requires less computational efforts, while it is more sensitive to noises and gradients of images, which has important impacts on the correction performance [18]. The mean difference for Sentinel 2B images corrected by ICM is about 0.5 pixels [19]. On the other side, NDM and FMM often provide higher geolocation accuracy for the corrected images than ICM. For example, the geolocation error of FengYun-3D (FY-3D) Microwave Radiation Imager (MWRI) images corrected by CIM is no more than 0.3 pixels [8,17]. However, the computational cost may be larger up to several orders of magnitude than with ICM [20], since NDM and FMM methods require to correct the satellite flight parameters based on a chain of coordinate transformation models leading from the antenna coordinate system to the geoid system [21,22].

In this study, we present a simplified CIM method (SCIM) to correct geolocation errors of satellite PMW images without implementing the complex coordinate transfers and demonstrate its usefulness by applying to FY-3D MWRI images. This paper is organized as follows. Section 2 describes study data and presents SCIM in detail. Section 3 shows the performance of SCIM for correcting the geolocation errors of FY-3D MWRI 89GHz horizontal polarization (h-pol) BT images. The discussion of the results is shown in Section 4 and conclusions in Section 5.

## 2. Data and Methods

### 2.1. FY-3D MWRI Images and Land/Sea Mask Data

The Chinese FengYun (FY) series of meteorological polar orbiting satellites were initiated in 1990s. There are four FY-3 satellites that operate in an afternoon orbit, a midmorning orbit, and a Sun-synchronous orbit. These satellites provide global observations for climate monitoring and analysis. FY-3D is the fourth unit of the second-generation Chinese polar orbiting meteorological satellites series launched on 15 November 2017. It is equipped with 10 remote sensing instruments which provide abundant information for surface features and atmospheric variables. The MWRI carried by FY-3D is a PMW sensor that scans the Earth conically with a zenith angle of 53.1° and a swath width of 1400 km. It provides measurements of radio PMW at 10.65, 18.7, 23.8, 36.5, and 89 GHz, each with horizontal and vertical polarization [23]. The effective field of view of MWRI at 89 GHz has a ground resolution of 9 km × 15 km. The FY-3D MWRI obtains 28 half-orbit swaths of BT observations of the Earth in one day (Figure 1). Due to the relatively high spatial resolution of the PMW images at 89 GHz, their BT data are widely used to retrieve land surface features, such as land surface temperature, sea ice concentration, snow water equivalent, snow depth,

etc. [24–26]. The 89.0 GHz h-pol FY-3D MWRI BT images of the first 3 days of every month from 2018 to 2019 are used in this study.

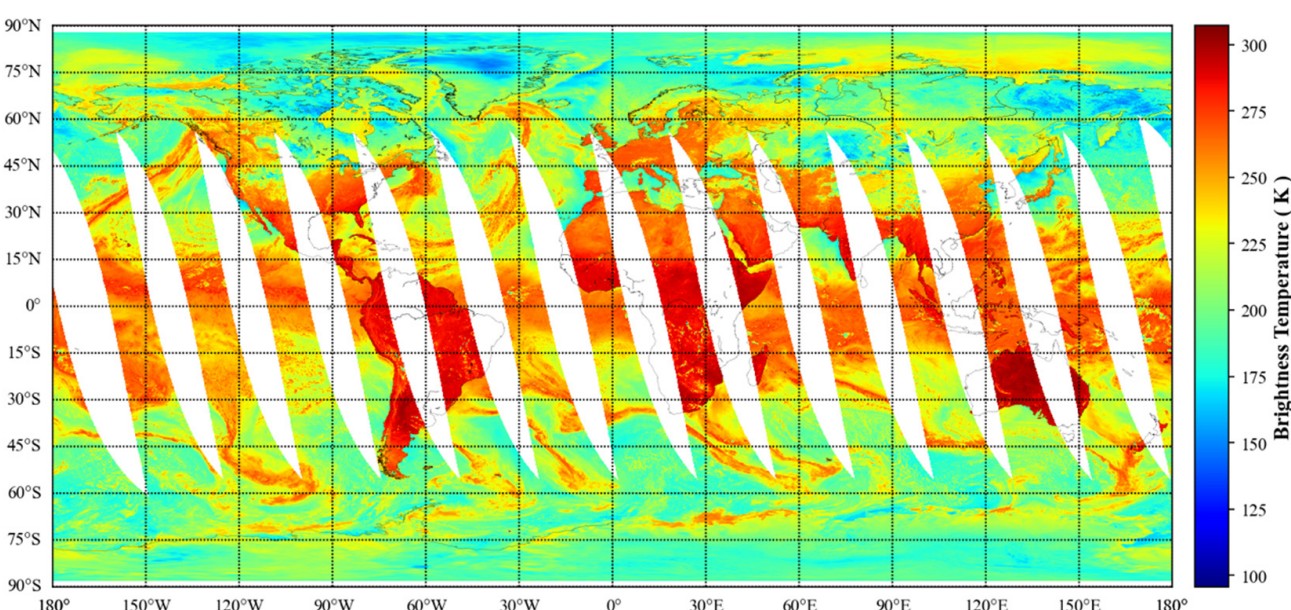

**Figure 1.** The FY-3D MWRI ascending brightness temperature on 1 January 2019.

In addition to the FY-3D MWRI images, two land/sea mask datasets are used in this study. One is the Global Self-consistent, Hierarchical, High-resolution Shoreline (GSHHS) provided by University of Hawai'i and NOAA Laboratory for Satellite Altimetry [27]. It was amalgamated from World Vector Shorelines, CIA World Data Bank II, and Atlas of the Cryosphere. The latest data files for version 2.3.7 were released on 15 June 2017. The other dataset is the LandSeaMask dataset provided by the National Satellite Meteorological Center (NSMC), China Meteorological Administration. NSMC transfers the GSHHS data from the ESRI vector to the LandSeaMask raster format with the same format and spatial resolution as the FY-3D MWRI images. LandSeaMask is used to correct the geolocation errors of FY-3D MWRI images, while GSHHS is used to assess the geolocation errors of the corrected FY-3D MWRI images.

### 2.2. The Simplified Coastline Inflection Method (SCIM)

In this study, SCIM corrects geolocation errors of MWRI images without coordinate transformations. SCIM is working in the swath coordinate system of the MWRI Level-1 images, i.e., with the directions along scan and cross scan as x and y axes. The method is organized in three steps (Figure 2). The first step is to find coastline inflection points from a MWRI image. Then, geolocation errors of the MWRI image in along the satellite flight direction (the along-track direction) and perpendicular to the satellite flight direction (the cross-track direction) are estimated. Finally, SCIM corrects the geolocation errors of the MWRI image in the along-track and the cross-track directions. The details of SCIM are described in the following.

#### 2.2.1. Finding Coastline Inflection Points in a MWRI Image

Land (shown as reddish pixels in Figure 3a) usually has higher BT than sea (bluish pixels in Figure 3a) in a MWRI image, resulting in large BT gradients appearing along the coastlines (Figure 3a). SCIM selects the pixels with largest BT gradient as coastline pixels in the MWRI image. The center points of the coastline pixels are the coastline inflection points.

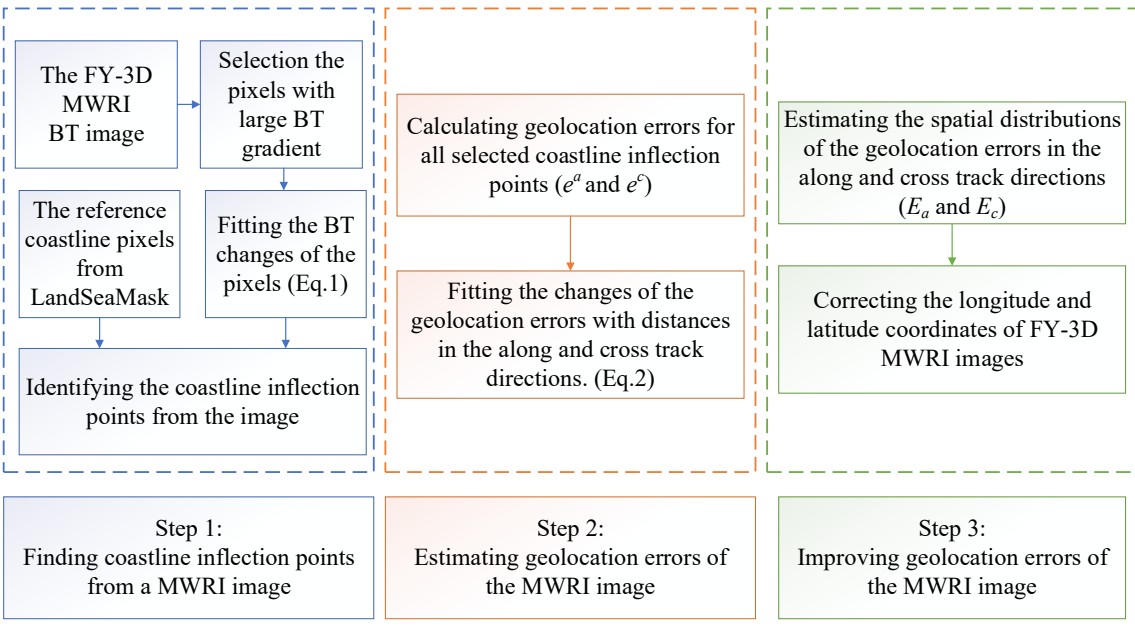

**Figure 2.** The flowchart of the SCIM geolocation correction for FY-3D MWRI data.

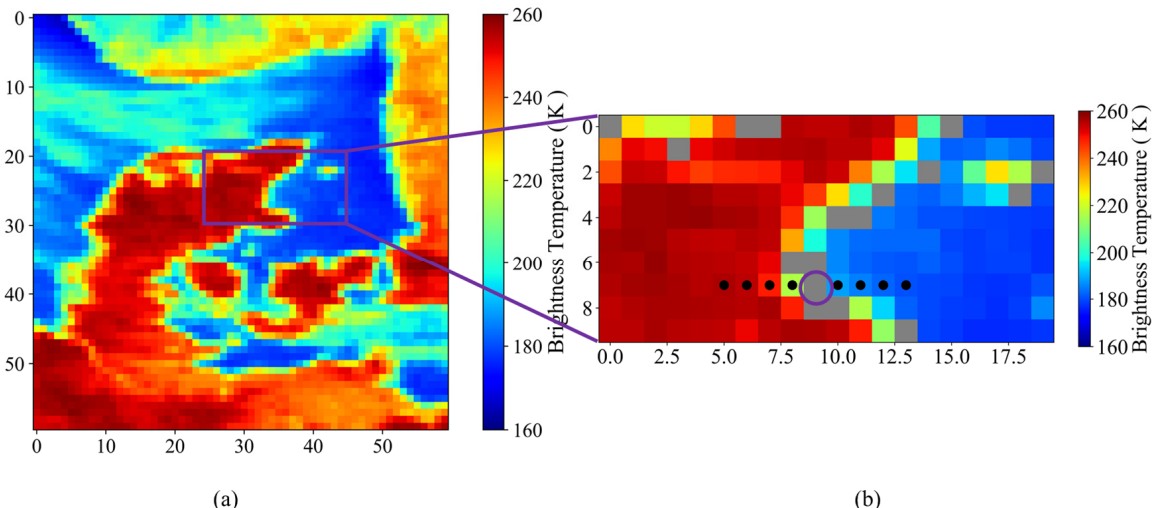

(a)                                                                                      (b)

**Figure 3.** The distribution of FY-3D MWRI brightness temperature of Denmark (**a**) acquired on 3 March 2018. (**b**) shows a zoom area in Aalborg Bugt, Denmark overlaid with the coastline point data. The highlighted gray pixels (**b**) represent the locations of the coastline pixels from LandSeaMask. The data are displayed in the swath coordinate system.

The details for finding coastline inflection points are shown as follows. According to previous studies, the geolocation error of the uncorrected MWRI image is less than four pixels [8]. We assume that the pixels with largest BT gradient would represent near the true coastline pixels. SCIM uses the coastline pixels from LandSeaMask as the true coastline pixels (gray pixels in Figure 3b). Then, SCIM takes BT values from the true coastline pixel (highlighted with the purple circle in Figure 3b) and its eight neighbor pixels along the direction of one of the two coordinates. Figure 3b shows an example with the neighbors taken in the cross-track direction. In the along-track direction, the procedure works similarly.

SCIM takes four pixels with relatively larger BT gradient from the eight pixels to find the pixel with the largest BT gradient. The BT values of the true coastline point (the blue point

in Figure 4a) and the four points (black points in Figure 4a) are shown in Figure 4a. The BT changes of these points can be plotted as Figure 4b. SCIM fits the polynomial function

$$y = \alpha \times x^3 + \beta \times x^2 + \gamma \times x + \delta \qquad (1)$$

to the BT values where $y$ represents the BT of the true coastline pixel and its neighbor pixels. $x$ represents the column numbers of these pixels ($x = 1,2,3,4$). The parameters $\alpha$, $\beta$, $\gamma$, and $\delta$ are determined by the fit. The point with the maximum of BT gradient (red point shown in Figure 4a,b) considered as the coastline inflection point, is obtained by calculating the first derivative of Equation (1). The distance from the coastline inflection point (the red point in Figure 4b) to the true coastline point (the blue point in Figure 4b) is the geolocation error for this pixel. SICM searches coastline inflection points in all rows (the along-track direction) and columns (the cross-track direction) of a MWRI image to get the geolocation errors for all coastline inflection points.

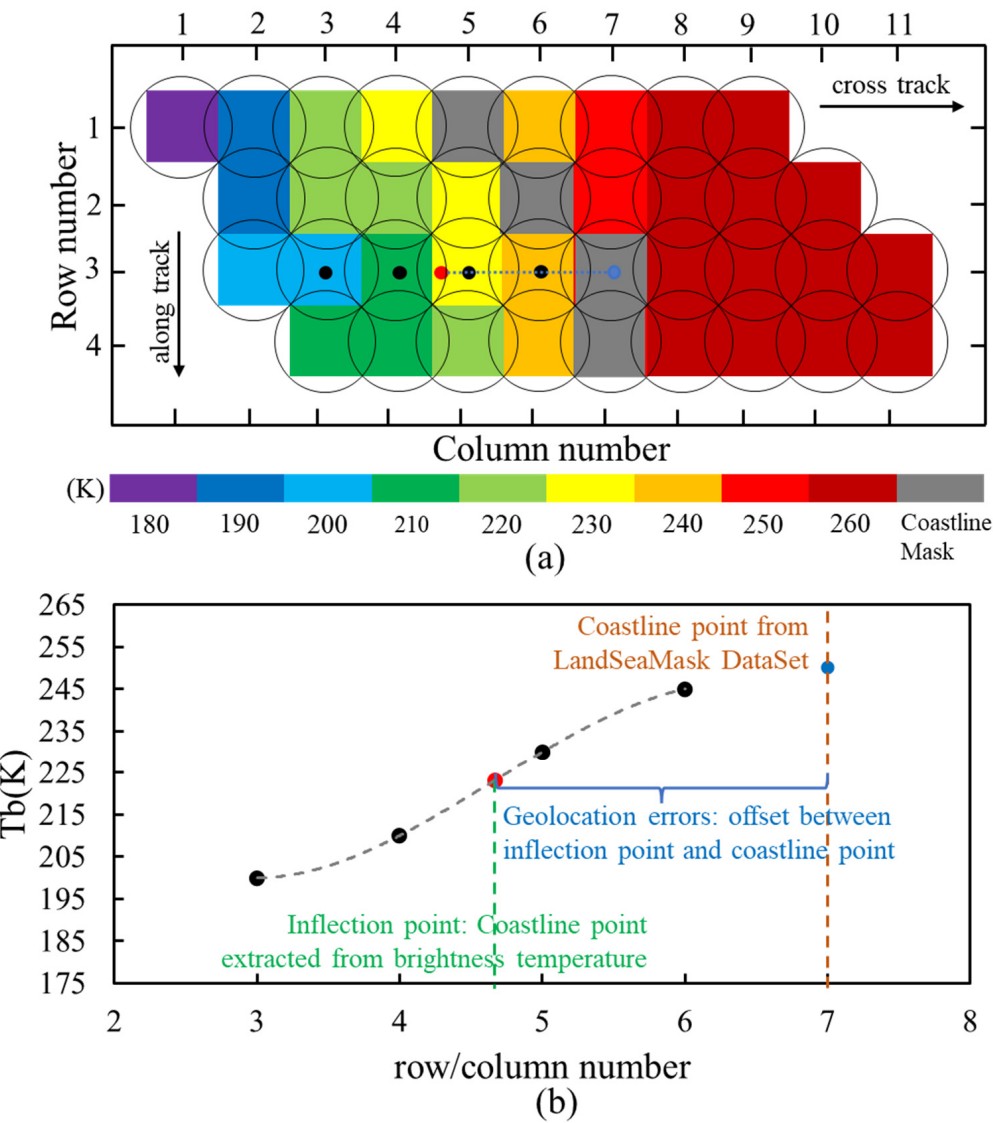

**Figure 4.** Diagram for finding coastline inflection points in the cross-track direction. The gray pixels are coastline pixels from LandSeaMask. The blue point is the center point of the coastline pixel, which represents the true coastline point. The red point is the coastline inflection point found by SICM. The colorful background in (**a**) represents changes of brightness temperature. (**b**) shows the BT changes of the coastline inflection points.

2.2.2. Estimating Geolocation Errors of a MWRI Image

The distance from a coastline inflection point found by SCIM and its corresponding true coastline point is the geolocation error of this pixel. We denote the distance as $e_{i,j}^a$ and $e_{i,j}^c$. Here and in the following, the subscripts $i$ and $j$ are row number and column number of the pixel, and the superscripts $a$ and $c$ represent the along-track and the cross-track directions.

To correct geolocation errors of a MWRI image, geolocation errors both in the along-track and the cross-track directions are needed. In this study, we explore the changes of geolocation errors ($e_{i,j}^a$ and $e_{i,j}^c$) of the coastline pixel with the distances from the coastline pixel to the scan line (in the along-track direction, the green line in Figure 5b) and the scan position (in the cross-track direction, the brown line in Figure 5b). We denote the distances from the coastline pixel $P(i,j)$ to the scan line and to the scan position as $d_{i,j}^a$ and $d_{i,j}^c$.

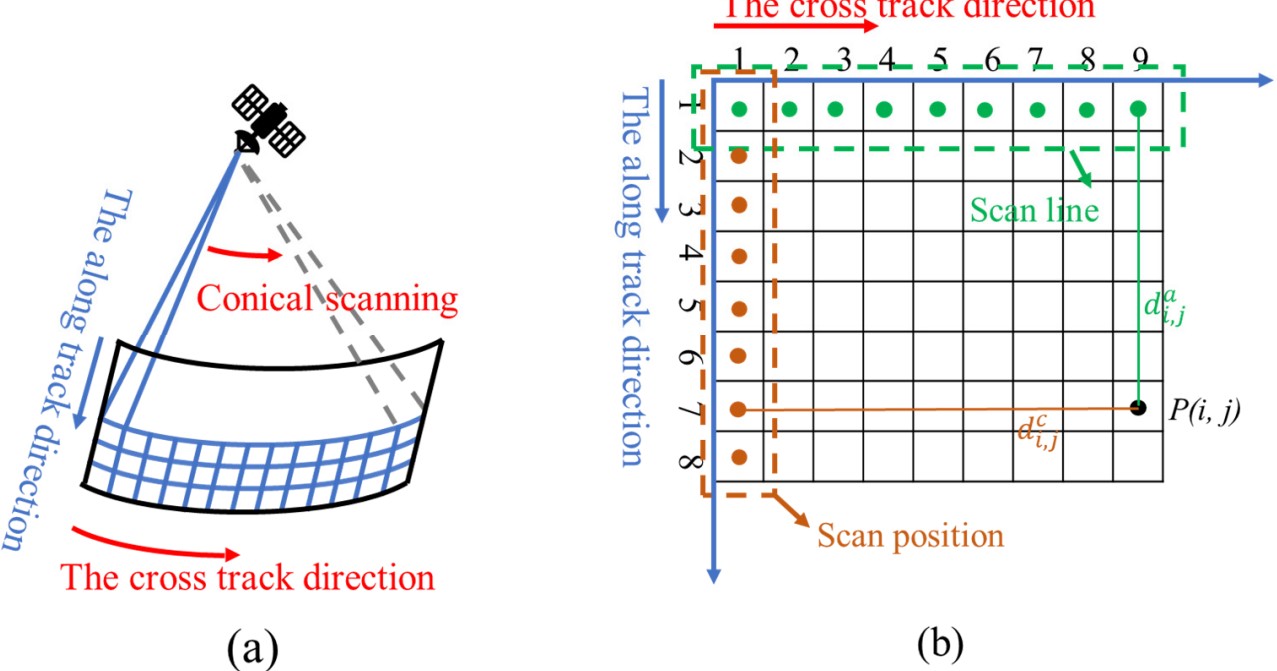

**Figure 5.** Conical scanning geometry of the MWRI instrument. (**a**) shows the scanning schematic of the FY-3D MWRI. (**b**) shows the schematic of FY-3D MWRI brightness temperature images. The red arrow points to the cross-track direction. The blue arrow points to the along-track direction.

Figure 6 shows the changes of geolocation errors ($e_{i,j}^a$ and $e_{i,j}^c$) for all coastline pixels and the distances ($d_{i,j}^a$ and $d_{i,j}^c$) between the pixels to the scan line and the scan position from a MWRI swath image obtained on the first day of odd-numbered months in 2018. The details of the MWRI images are shown in Table 1. As shown in Figure 6, positive and negative geolocation errors are detected in the descending and ascending MWRI image, respectively. The geolocation errors mostly range from 0 to 2 pixels for the ascending images and −2 to 0 pixels for the descending images. One pixel corresponds to 9 km in the along-track direction and to 15 km in the cross-track direction. The changes of geolocation errors with the distances $d_{i,j}^a$ and $d_{i,j}^c$ are quite different in the along-track and the cross-directions. There is a clear positive linear relationship between the geolocation errors $e_{i,j}^c$ and the distances $d_{i,j}^c$ in the cross-track direction, while the geolocation errors in the along-track direction are independent of the distances $d_{i,j}^a$. The correlation coefficients for the geolocation errors and the distances ($d_{i,j}^a$ and $d_{i,j}^c$) also support this point (Table 1). In addition, the geolocation errors in the cross-track direction are slightly larger than the errors in the along-track direction. The geolocation errors in the along-track direction are

usually no more than 2 pixels, while the geolocation errors in the cross-track direction reach 3 pixels.

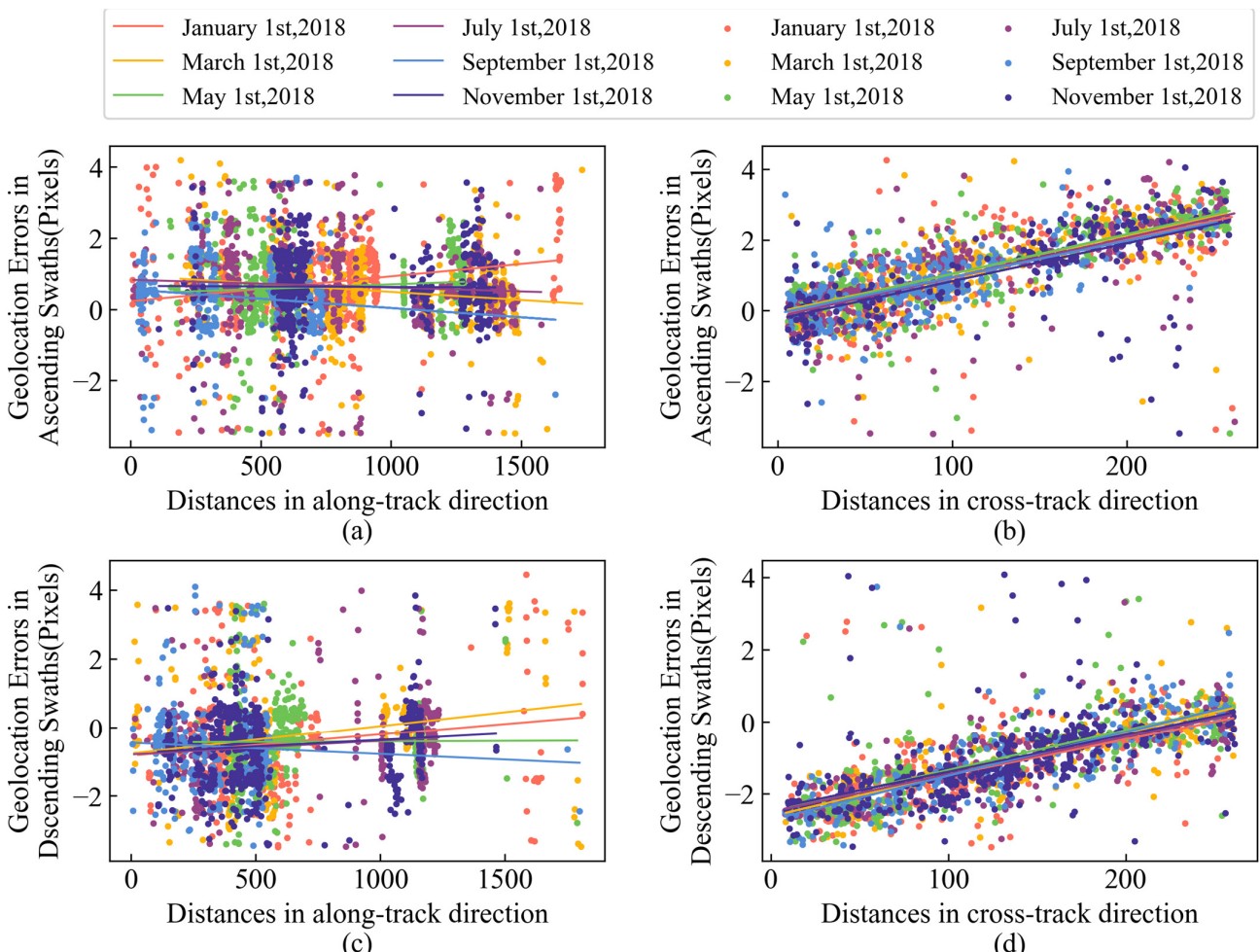

**Figure 6.** Geolocation errors of the coastline pixels found by SCIM for ascending (**a**,**b**) and descending (**c**,**d**) images on different dates. Horizontal axis are distances from the pixels to the scan line and the scan position in the cross-track and the along-track direction ($d_{i,j}^a$ and $d_{i,j}^c$). Colored scatters are geolocation errors. Different colors represent different dates. The colored lines are linear fits between the geolocation errors and the distances. All distances (x axes, all diagrams) and geolocation errors (y axes, all diagrams) are given in pixels.

**Table 1.** The correlation coefficients of the geolocation errors $e_{i,j}^a$, $e_{i,j}^c$ and the distance $d_{i,j}^a$, $d_{i,j}^c$ in the along-track and cross-track direction. Each line represents the analysis of one full half orbit starting at the indicated UTC time.

| Date | UTC Time | Image | Correlation Coefficient (Along-Track Direction) | p Value (Along-Track Direction) | Correlation Coefficient (Cross-Track Direction) | p Value (Cross-Track Direction) | Alpha (α) | Beta (β) |
|---|---|---|---|---|---|---|---|---|
| 1 January 2018 | 09:50 | ascending | 0.05 | <0.01 | 0.44 | <0.01 | 0.010 | −0.15 |
| 1 March 2018 | 18:03 | ascending | 0.03 | <0.01 | 0.56 | <0.01 | 0.011 | −0.04 |
| 1 May 2018 | 12:05 | ascending | 0.01 | <0.01 | 0.58 | <0.01 | 0.011 | −0.08 |
| 1 July 2018 | 17:57 | ascending | 0.01 | <0.01 | 0.45 | <0.01 | 0.011 | −0.18 |
| 1 September 2018 | 20:07 | ascending | 0.02 | <0.01 | 0.43 | <0.01 | 0.010 | −0.09 |
| 1 November 2018 | 04:00 | ascending | 0.00 | <0.01 | 0.45 | <0.01 | 0.011 | −0.25 |
| 1 January 2018 | 02:13 | descending | 0.02 | <0.01 | 0.43 | <0.01 | 0.010 | −2.48 |
| 1 March 2018 | 23:58 | descending | 0.05 | <0.01 | 0.59 | <0.01 | 0.012 | −2.63 |
| 1 May 2018 | 23:05 | descending | 0.00 | <0.01 | 0.47 | <0.01 | 0.011 | −2.47 |
| 1 July 2018 | 15:25 | descending | 0.03 | <0.01 | 0.48 | <0.01 | 0.010 | −2.51 |
| 1 September 2018 | 00:40 | descending | 0.00 | <0.01 | <0.01 | <0.01 | 0.012 | −2.73 |
| 1 November 2018 | 16:41 | descending | 0.02 | <0.01 | 0.34 | <0.01 | 0.010 | −2.43 |
| Mean | - | - | 0.02 | | 0.49 | | - | - |

Based on these results, SCIM estimates the geolocation errors for a MWRI image in the along-track and the cross-track directions. We use the mean value of the geolocation errors of all coastline pixels found by SCIM as the geolocation error in the along-track direction ($e^a$) for the image. For the cross-track direction, we use the linear function

$$e^c = \alpha \times x + \beta \tag{2}$$

to describe the relationship between geolocation errors and the distance $d_{i,j}^c$. $e^c$ and $x$ present the geolocation errors in the cross-track direction and the distance $d_{i,j}^c$. $\alpha$ and $\beta$ are underestimated by the fit. The fitted $\alpha$ and $\beta$ are shown in Table 1.

### 2.2.3. Correction of Geolocation Errors

Once $e^a$ and $e^c$ are determined, SCIM can be used to correct the geolocation. Two geolocation error matrixes are obtained to describe the spatial distributions of the geolocation errors in the along/cross-track direction. One is the geolocation errors matrix ($E_a$) representing the spatial distribution of the geolocation errors in the along-track direction, while the other is the geolocation errors matrix ($E_c$) representing the spatial distribution of the geolocation errors in the cross-track direction. The two geolocation error matrixes have the same size as the MWRI image. For $E_a$, all elements equal to $e^a$, while the elements in $E_c$ are estimated by Equation (2). Since the unit of $E_a$ and $E_c$ is pixel, we need to change the unit to geographic coordinates for correcting the geolocation. A bilinear function is used to implement the transform for longitudes and latitudes. We firstly use two bilinear functions to fit the relationships between column and row numbers of the MWRI image and longitudes or latitudes of the image in swath coordinate system, respectively. The bilinear functions can be expressed as B(x,y,z), where x, y are column and row numbers of the MWRI image respectively. z is longitude or latitude matrixes of the image. The longitude and latitude matrixes are provided by NSMC for each MWRI image. They have the same size as the image. Then, $E_a$ and $E_c$ are respectively added to x and y to get updated column and row numbers labeled as $\hat{x}$ and $\hat{y}$. Finally, the corrected longitudes and latitudes are calculated by using the bilinear functions. It can be expressed as

$$\hat{z} = B(\hat{x}, \hat{y}) \tag{3}$$

where $B$ is the fitted bilinear function, $\hat{z}$ is the corrected longitudes and latitudes. Since $e_a$ and $e_c$ are different from image to image, it is necessary to apply SCIM to every single MWRI image (half orbit) for correcting their geolocation errors.

### 2.3. Evaluation Statistics

Root mean square error (*RMSE*) and error reduction (*ER*) are used to evaluate the geolocation correction performances of SCIM on the complete half-orbits. *RMSE* measures the distance between the coastline inflection points extracted from the images and the points from GSHHS. A large *RMSE* means the geolocation error is large. *RMSE* can be calculated as the following equation.

$$RMSE = \left( \sum_{n=1}^{N} (x_n - G_n)^2 / N \right)^{1/2} \tag{4}$$

Here $x_n$ is one of the along-track or cross-track coordinates of the coastline inflection points from the images. $G_n$ is the corresponding along-track or cross-track coordinate of the coastline inflection points from GSHHS. $N$ is the number of the coastline inflection points. The *RMSE*s are calculated in the along-track and cross-track direction. The error reduction rate (*ER*) is defined as:

$$ER(\%) = (1 - RMSE_{cor} / RMSE_{uncor}) \times 100 \tag{5}$$

where $RMSE_{cor}$ and $RMSE_{un\text{-}cor}$ are $RMSE$ for the corrected and uncorrected BT images. A large $ER$ value means the method better reduces the geolocation errors of the image.

## 3. Results

### 3.1. Evaluation of Geolocation Accuracy before and after Geolocation Correction

The geolocation errors obtained in the first three days of every month from 2018 to 2019 are corrected by SCIM. The average $RMSE$s and $ER$s for the uncorrected and corrected images are shown as Table 2. The $RMSE$s are reduced from 0.575 resp. 1.098 to 0.145 resp. 0.149 pixels in the along/cross-track direction. The $ER$s are 74.78% and 86.39% in the along-track direction and the cross-track direction, respectively. These results indicate that SCIM has the ability to improve geolocation accuracy of MWRI images. We also use NCC to correct the geolocation errors of the images. The geolocation errors of the images corrected by NCC are reduced. The average geolocation errors of the image corrected by NCC is 0.151 pixels in the along-track direction, which is slightly higher than that of SCIM, while NCC geolocation errors in the cross-track direction is 0.498 pixels, which is much higher than that of SCIM (Table 2)

**Table 2.** Comparisons of the geolocation errors of the MWRI brightness temperature images corrected by SCIM and NCC.

|  | Along-Track (Unit: Pixel) | | | Cross-Track (Unit: Pixel) | | |
|---|---|---|---|---|---|---|
|  | Before Correction | After Correction | Error Reduction | Before Correction | After Correction | Error Reduction |
| SCIM | 0.575 | 0.145 | 74.78% | 1.098 | 0.149 | 86.43% |
| NCC | 0.575 | 0.151 | 73.74% | 1.098 | 0.498 | 54.64% |

Since large BT gradients are easily found in the coastline regions in a PMW image, the distance between the gradients regions and the coastline regions is usually used to illustrate the geolocation errors of PMW images. Figure 7 shows the distributions of the uncorrected and corrected MWRI images in four coastline regions, including Severnaya Zemlya, Denmark, Australia, and Persian Gulf. Figure 7 also shows the coastline from GSHHS (black lines). Before the geolocation correction, the pixels with large brightness temperature gradients in the MWRI images are far from the GSHHS coastlines. Some pixels with high BT distribute in the oceans (Figure 7c), while some pixels with low BT locate in the lands (Figure 7g). The distances between pixels with large BT gradients to the GSHHS coastline data are significantly reduced when SCIM has been applied to correct the geolocation errors (Figure 7b,d,f,h). These results confirm and illustrate the reduction of the geolocation errors of MWRI images by SCIM.

### 3.2. Comparisons of the Brightness Temperature Differences before and after Geolocation Correction

The BT differences between ascending and descending images on the same day are often used in evaluating geolocation accuracy [14]. Lower BT differences mean better geolocation accuracy [5]. The FY-3D MWRI images obtained in the first three days of every month from 2018 to 2019 are corrected by SCIM. The mean BT differences of the uncorrected and corrected images are shown in Figure 8. The mean BT difference of the uncorrected images is 25.54 K. It is reduced to 16.83 K for the corrected image, by again confirming the geolocation improvement by the Table 3. Table 3 shows the mean BT differences for the four regions shown in Figure 7 The mean BT differences of the uncorrected MWRI images range from 19.66 K to 26.72 K, while the differences for the corrected images range from 11.90 K to 16.80 K. The mean BT differences are reduced by 31% when SCIM is used. The reductions of BT difference between ascending and descending MWRI images imply that the geolocation errors are significantly improved. Note that a reduction of the difference to 0 K cannot be expected because of the temperature difference occurring in the time between ascending and descending overflights.

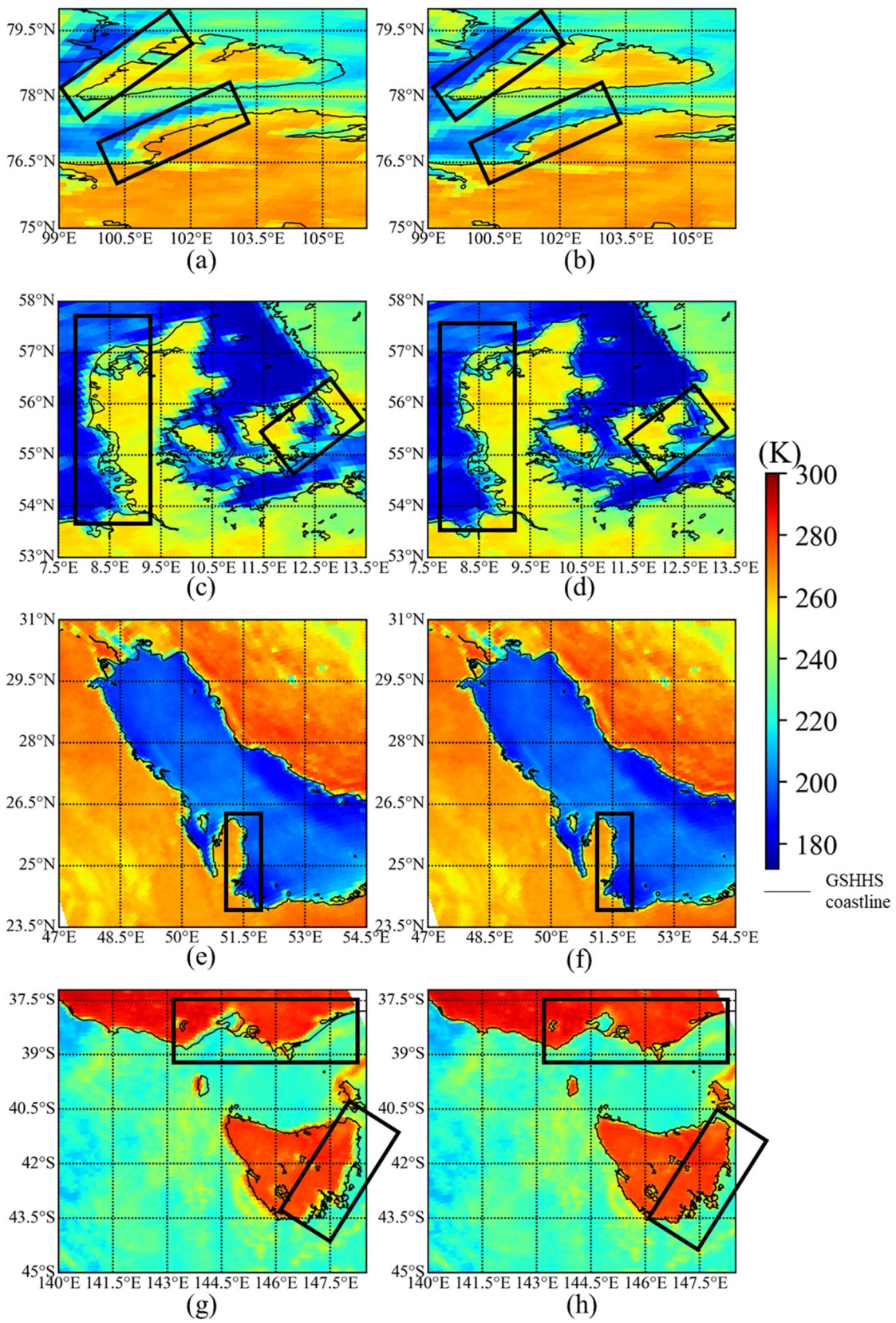

**Figure 7.** MWRI 89 GHz h-pol brightness temperature maps of four regions uncorrected (**a**,**c**,**e**,**g**) and corrected by SCIM (**b**,**d**,**f**,**h**). From top to bottom, the regions are Severnaya Zemlya (**a**,**b**), Denmark (**c**,**d**), the Persian Gulf (**e**,**f**), and Australia (**g**,**h**). The black boxes show the BT comparisons of some coastline regions between uncorrected and corrected FY-3D MWRI images.

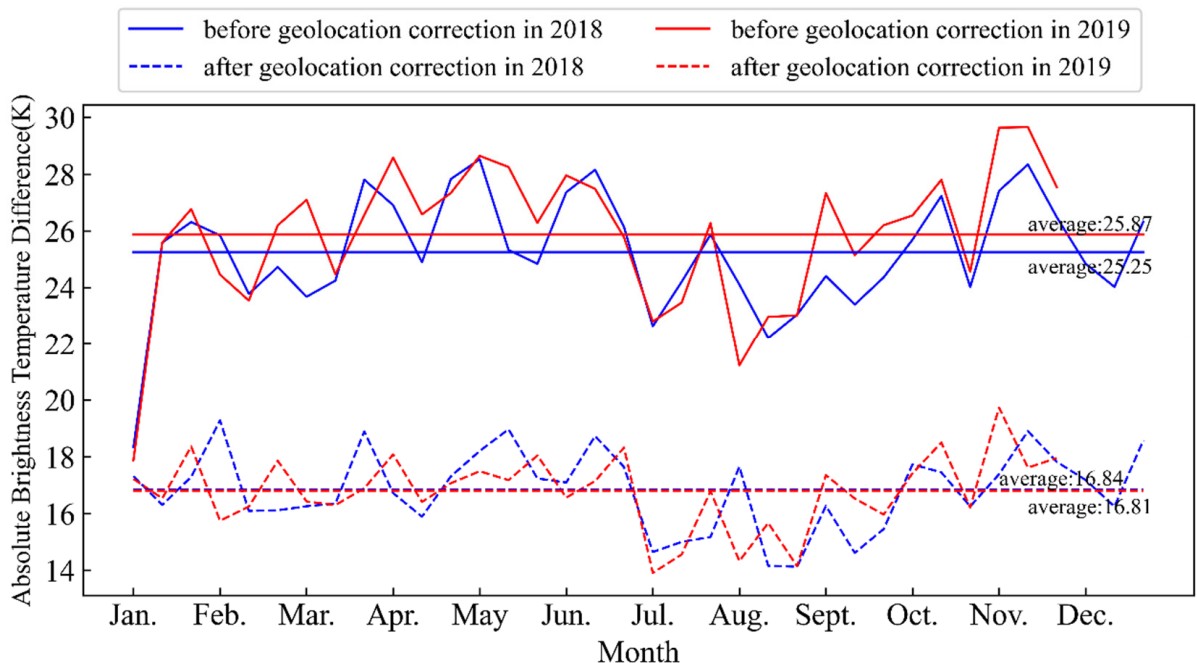

**Figure 8.** The brightness temperature differences of the MWRI images before and after geolocation correction by SCIM in the first three days of every month from 2018 to 2019.

**Table 3.** Mean brightness temperature differences between MWRI ascending and descending images in the first three days of every month from 2018 to 2019.

|  | Before Correction | After Correction | Decline |
|---|---|---|---|
| Severnaya Zemlya | 19.66 | 13.39 | 31.88% |
| Denmark | 23.43 | 11.90 | 49.23% |
| Australia | 25.59 | 15.32 | 40.15% |
| The Persian Gulf | 26.72 | 16.80 | 37.11% |

*3.3. Comparisons of Accuracy of Sea Ice Concentration Retrieved from MWRI Images before and after Geolocation Correction*

Sea ice concentration (SIC) is an essential variable for monitoring Arctic sea ice changes. Satellite PMW data are commonly used for sea ice concentration retrievals [28–31]. Since the geolocation error of PMW images is an error source for sea ice concentration products [5], the accuracy of SIC retrieved by uncorrected and corrected images also can illustrate the performance of the correction method.

In this study, we retrieved daily Arctic SIC by using uncorrected and corrected MWRI images in 2019. Due to the high spatial resolution (250 m) of the MODIS surface reflectance product (MOD09), it is usually used to evaluate SIC products retrieved from PMW data [31,32]. The SICs estimated by the MOIDS images are compared with the SICs retrieved from the corresponding MWRI images. Figure 9 shows the differences of the SICs retrieved by the uncorrected and corrected MWRI images on 22 March (the maximum sea ice cover appearing in March) and 3 September (the minimum sea ice cover appearing in September) in 2019. The black boxes in Figure 9 are the sample regions with cloud free MODIS images used. Obvious differences of the SICs appear near the sea ice margin regions and the coastline regions. The differences range from −20% to 20%. By comparing with the SICs retrieved from MOD09, the images corrected by SCIM have better performance on estimating Arctic SICs than the uncorrected images (Table 4). The mean $RMSE$/MAE is reduced from 13.67%/8.92% to 10.22%/6.85%. The mean correlation coefficient increases from 0.91 to 0.95. Figure 10 shows the spatial differences between the SICs from the MWRI images and the MOD09 images in the three sample regions. The differences between

resulting from the uncorrected in coastline regions are shown in Figure 10a,c,e, and those resulting from the corrected images are also found in Figure 10b,d,e. The differences based on the corrected images are significantly smaller than those based on the uncorrected ones. All these results indicate that SCIM is well suited for improving the geolocation errors of FY-3D MWRI images.

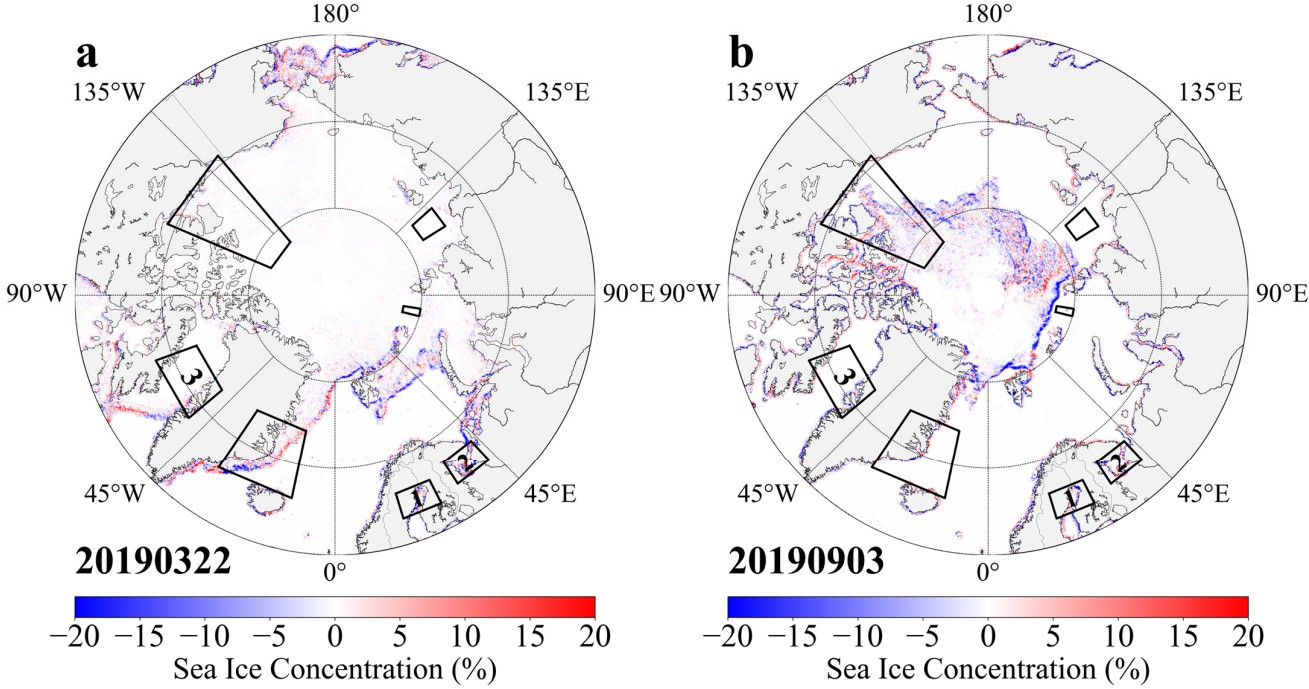

**Figure 9.** SIC differences retrieved by the uncorrected and corrected MWRI images in 22 March (**a**) and 3 September 2019 (**b**). The black boxes are the sample regions with cloud free MODIS images.

**Table 4.** Comparisons of the SICs from the uncorrected and corrected images. MAE and R are mean absolute error and correlation coefficient, respectively.

| Image Acquiring Dates | RMSE | | MAE | | R | |
|---|---|---|---|---|---|---|
| | Original | Corrected | Original | Corrected | Original | Corrected |
| 17 February 2019 | 20.70 | 15.16 | 15.79 | 11.97 | 0.84 | 0.93 |
| 22 March 2019 | 18.58 | 14.34 | 14.39 | 11.45 | 0.85 | 0.92 |
| 29 April 2019 | 10.02 | 6.96 | 5.55 | 4.18 | 0.96 | 0.98 |
| 30 May 2019 | 10.20 | 7.41 | 5.15 | 4.12 | 0.97 | 0.98 |
| 6 June 2019 | 8.80 | 6.41 | 3.94 | 2.97 | 0.98 | 0.99 |
| 11 July 2019 | 17.00 | 12.02 | 11.11 | 7.48 | 0.80 | 0.90 |
| 3 September 2019 | 10.40 | 9.26 | 6.54 | 5.78 | 0.97 | 0.98 |
| Mean | 13.67 | 10.22 | 8.92 | 6.85 | 0.91 | 0.95 |

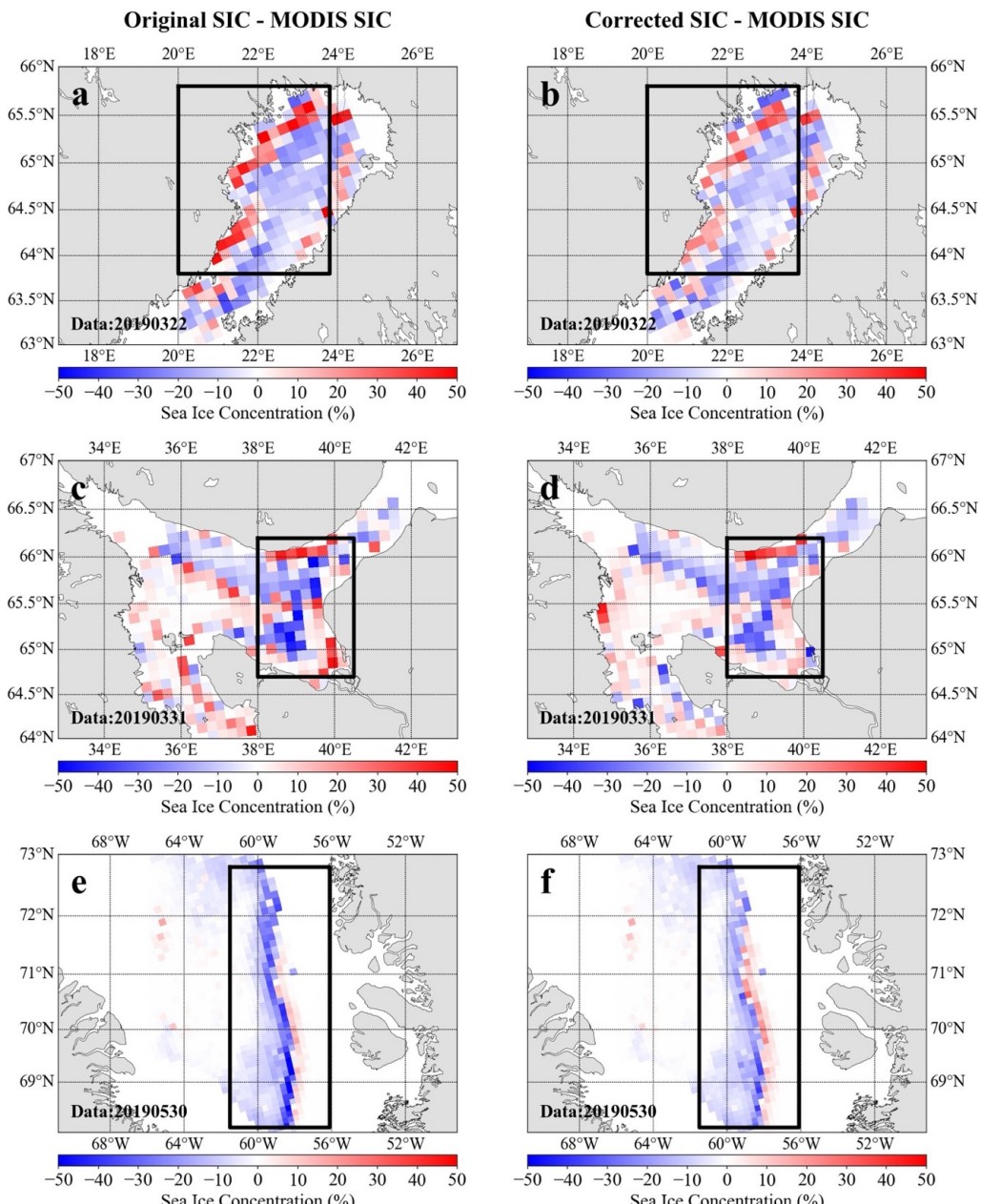

**Figure 10.** The spatial differences between the SICs from the MWRI images and the MOD09 images in three sample regions. These sample regions are shown as the black boxes labeled by numbers in Figure 9. From top to bottom, the regions are Gulf of Bothnia (**a**,**b**), White Sea (**c**,**d**), and Davis Strait (**e**,**f**).

## 4. Discussion

In this study, SCIM is proposed to correct geolocation errors of satellite PMW images. It has been applied to correct the geolocation errors of FY-3D MWRI images. The geolocation errors of the corrected images are reduced by 74.78% in the along-track direction and by 86.43% in the cross-track direction. By comparing with NCC, the geolocation error reductions by SCIM are stronger than the reductions of the images corrected by NCC. All these results indicate that SCIM effectively improves the geolocation errors of the FY-3D MWRI images.

Several CIM methods have been developed to correct the geolocation errors of MWRI images. For example, Tang and Li et al. used CIM to correct FY-3C/D MWRI data [6,16,17]. All these previous studies have made outstanding contributions for exploring the way

to correct the geolocation errors of FY-3C/D MWRI images. Their methods corrected the geolocation errors of FY-3D MWRI images by improving the satellite flight parameters of the FY-3D satellite. Our study is an exploration for correcting geolocation errors of FY-3D MWRI images in the image coordinate system without correcting the satellite flight parameters. According to their studies, the geolocation error reductions (*ER*) range from 70% to 80% in the along-track direction and from 70% to 90% in the cross-track direction. The *ER* of SCIM is in the range of the reductions of existing methods. However, computational cost for the geolocation correction methods based on improving satellite flight parameters is much higher than the methods based on the swath coordinate system, since they are often required to optimize satellite flight parameters and involve complex coordinate system transforms from the antenna coordinate system to the geoid system [33,34]. The SCIM requires 62 s to correct a FY-3D MWRI image on a personal computer with Intel(R)Core(TM)i7-8550U CPU@1.80GHz and 16 GB RAM.

The advantages of SCIM are fast computation and that the full set of orbit parameters which may not be accessible to all users, are not needed. On the other hand, geolocation based on the complete orbit parameters can provided a first guess needed by SCIM. Moreover, we expect that orbit parameter-based geolocation yields the most accurate results if these parameters are known with sufficient accuracy. Since SCIM is simple and has a good performance for geolocation correction, it can be a useful method for users who get satellite based passive microwave images with geolocation errors, but without satellite flight parameters and coordinate system transform.

Our result shows a linear increasing of the geolocation errors with the distance increasing ($d_{i,j}^c$) in the cross-track direction, while the geolocation errors are independent on the distance ($d_{i,j}^a$) in the along-track direction. Since the geolocation errors in the cross-track and along-track direction are different, SCIM uses different methods to describe their spatial distributions. A constant error is used to represent the distribution of the geolocation errors in the along-track direction, while a linear function is used for the cross-track direction. For the cross-track direction, the linear function has a good performance for describing the changes, reducing the error by nearly 90%. However, in the along-track direction, the geolocation errors are practically uncorrelated (Figure 5a,c) so that SCIM only estimates a constant to correct for the system bias. This results in a lower error reduction (no more than 80%) for the along-track direction. We suggest that the constant used to correct geolocation errors in the along-track direction is only a simplified description for the spatial distribution of the errors. It is necessary to explore the error distributions along-track in different regions in order to develop a more accurate SCIM in the along-track direction.

## 5. Conclusions

In this study, SCIM is proposed to correct for geolocation errors in satellite PMW images without implementing complex coordinate system transforms. It takes advantage of the large brightness temperatures gradients between land and sea to extract coastline inflection points. The extracted coastline inflection points are compared with the LandSea-Mask dataset to estimate the geolocation errors in the swath coordinate system (along track and cross track) and to obtain distributions of the geolocation errors. Finally, the SCIM uses the distributions of geolocation errors to recalculate corrected lat/lon coordinates for all pixels in the swath image.

FY-3D MWRI 89-GHz h-pol brightness temperature images of the first three days every month from 2018 to 2019 are collected and corrected by the SCIM. The main results are: (1) SCIM reduces the geolocation errors from 0.575/1.098 to 0.145/0.149 pixels in the along/cross-track direction, respectively, corresponding to relative error reductions of 74.78% and 86.43%. (2) The mean brightness temperature differences between ascending and descending FY-3D MWRI images are reduced by 34.1%, which demonstrates the good correction performances of the SCIM. Note that a reduction of the difference to 0 K cannot be expected because of the temperature difference occurring in the between ascending and descending overflight. (3) The uncorrected and corrected MWRI images are used

to retrieve daily Arctic SICs and to be compared with the SICs estimated by the MOD09 images. The mean *RMSE*/MAE of SICs is reduced from 13.67%/8.92% to 10.22%/6.85% when the corrected images are used, and the mean correlation coefficient also increases from 0.91 to 0.95.

All these results suggest that SCIM has ability to improve the geolocation errors of FY-3 MWRI images significantly.

**Author Contributions:** Conceptualization, X.C. and G.H.; methodology, G.H. and Z.C. (Zhuoqi Chen); software, J.X.; validation, J.X.; formal analysis, Z.C. (Zhuoqi Chen); investigation, J.X.; resources, J.X.; data curation, J.X.; writing—original draft preparation, Z.C. (Zhuoqi Chen); writing—review and editing, Z.C. (Zhaohui Chi), L.Y., S.W. and F.H.; visualization, J.X.; supervision, X.C. and G.H.; project administration, X.C.; funding acquisition, X.C. All authors have read and agreed to the published version of the manuscript.

**Funding:** This work was supported by the Guangdong Basic and Applied Basic Research Foundation (2021B1515020032), the National Key Research and Development Program of China (2019YFC1509104) and the National Science Fund for Distinguished Young Scholars (41925027).

**Data Availability Statement:** Not applicable.

**Conflicts of Interest:** The authors declare no conflict of interest.

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
