# Peer review of "A Simplified Coastline Inflection Method for Correcting Geolocation Errors in FengYun-3D Microwave Radiation Imager Images"

_remotesensing, doi:10.3390/rs15030813_

Round 1

Reviewer 1 Report

Authors in this research article have presented and investigated A Simplified Coastline Inflection Method for Correcting Geo-location Errors in FengYun-3D Microwave Radiation Imager Images. The topic and concept of the paper are interesting and it includes promising results. Prior to final acceptance recommendation the authors are encouraged to address the following comments.

1.      Explain more about “FY-3D” in the text of the article.

2.      Its language needs some minor modifications.

3.      Authors can provide a comparison of their proposed work with previous articles in a table.

4.      What is the difference between the Fig. 7 form of e and f?

5.      The authors have used the term antenna on pages 2 and 16. It is better to mention appropriate references of the antennas used for their work. The following may be helpful.

“Substrate Integrated Waveguide Leaky Wave Antenna with circular polarization and improvement of the scan angle”

“Magnetically Scannable Slotted Waveguide Antenna based on the Ferrite with Gain Enhancement”

“HIGH GAIN AND WIDEBAND MULTI-STACK MULTILAYER ANISOTROPIC DIELECTRIC ANTENNA”

“Design of a 1*4 Microstrip Antenna Array on the Human Thigh with Gain Enhancement”

Reviewer 2 Report

This manuscript does an interesting demonstration on a Simplified Coastline Inflection Method for Correcting Geo-location Errors in FengYun-3D Microwave Radiation Imager Images. Introduction is good. Very interesting presentation and very good data analysis. Convolutional neural networks have been broadly applied in hyperspectral image classification and have demonstrated impressive performance in the past decade. However, still some challenging problems still exist which can be covered in this paper. Passive microwave (PMW) sensors are popularly applied to earth observations. The satellite PMW radiometer data sometimes have errors in geolocation. Therefore, coastline inflection methods (CIMs) are widely used to improve geolocation errors of PMW images. However, they commonly require accuracy satellite flight parameters, which are difficult for obtaining by users. In this study, authors present a simplified coastline inflection method (SCIM) to correct the geolocation errors without demanding for the satellite flight parameters. During the reading of the manuscript, the following questions and comments came to my mind and I would like to ask the authors to comment on them:

1-         Please check the file attached for details comments

As the article provides a good body of work, I find that it has the sufficient quality to be published in present state.

Reviewer 3 Report

On the face of it, this seems to be an interesting paper discussing a simplified coastline inflection method for geolocation correction in microwave images. The study is generally well-designed with a fair evaluation of the proposed approach in terms of the performance and applications. However, the paper can be better organized with more details to emphasize the main focus and elucidate the key information. A proofreading of the format and grammar is also suggested to clarify some confusing argumentations and expressions in the paper. Specific comments are listed below:

- Page 2 Line 72-73, this statement is ungrounded and needs more elaboration. Based on what dataset is this mean difference calculated from?

- Page 3 Line 100, using the first three days of each month to develop this approach does not seem to be able to represent the changes of ground conditions, especially some periodic changes. What is the specific reason that these three days were used?

- Page 4 Line 104-112, it would be helpful to include some more background information concerning these two reference datasets used for approach development. How did the datasets develop? Approximately, what is the unresolvable error from using these datasets as inputs?

- Page 6 Line 191, how significant is the linear relationship?

- Page 8 Figure 6, this figure is hard to read. There is too much overlapping among different layers in these scatter plots. Breaking down into subplots and converting to density scatter plots might be better for the audience to interpret the figure. Please also use a more distinguishable color palette for the dots and lines.

- Page 13 Section 3.3, this section loosely fit the topic of this paper. It is hard to find a clear and concise conclusion about the applications of the results. It is suggested to reorganize the paragraph, abbreviate the information, and highlight the key message the authors trying to deliver.

- Page 16 Line 377-379, what is the performance of other correction methods in terms of efficiency? As the proposed approach was claimed to be simple and require less computation, how much improvement this approach achieved when compared with other approaches?

Round 2

Reviewer 1 Report

The authors have carefully responded to the referees' comments and this article can be published in this journal